# The Effects of Motivational Interviews About Activities of Daily Living on Physical Adjustment and Quality of Life in Elderly Total Knee Arthroplasty Patients: A Randomised-Controlled Trial

**DOI:** 10.3390/healthcare12232472

**Published:** 2024-12-06

**Authors:** Gizem Kubat Bakir, Sonay Göktas

**Affiliations:** 1School of Nursing, Maltepe University, Istanbul 34857, Turkey; 2Hamidiye Faculty of Nursing, University of Health Sciences, Istanbul 34668, Turkey; sonay.goktas@sbu.edu.tr

**Keywords:** motivational interview, quality of life, physical adjustment, activities of daily living

## Abstract

In old age, knee osteoarthritis is a common disease that reduces mobility. Total knee arthroplasty (TKA) is, in fact, a very important surgery to treat severe knee osteoarthritis. This study aimed to analyse the effect of motivational interviewing (MI) on physical adjustment and quality of life among old patients after TKA. Self-assessment forms were applied before and after the intervention using the functional assessment form and SF-36 Quality of Life Questionnaire in both groups: the intervention group who received MI targeted at daily living activities and the control group who received usual care within a randomised controlled trial that included 70 participants. The results we obtained showed significantly higher functional capacity scores and QOLs among those who underwent MI than those who did not have this additional support, while showing strong adjusted mean differences between two interventions that indicate this effect size difference. There was a notable increase in SF-36 scores from 51.14 to 85.77 which was much higher than the control’s rise from 45.97 to 59.46. Therefore, these findings suggest that many elderly people can greatly improve their health status after TKA with MI as it offers an opportunity for effective recovery during the post-operative period, especially among older adults. Therefore, it can be used as an efficient method included in standard routines after operations so that results are improved and patients’ satisfaction levels also increased simultaneously too.

## 1. Introduction

Old age is marked by frailty, excessive dependence on others, decreased ability to adapt to stressors outside the body, and a reduced resistance against them [1,2]. At this stage, people may begin to experience conditions such as rheumatoid arthritis, osteoporosis, and osteoarthritis (OA), among other degenerative joint diseases, or falls-related injuries such as pelvic fractures [3]. Knee osteoarthritis is a common disease that significantly reduces mobility in old age, affecting approximately 16.0% of people aged 15 years or older and 22.9% of those aged 40 years or older [4]. Total knee arthroplasty (TKA) is recommended for individuals suffering from severe OA that is unresponsive to other treatments.

Despite the benefits of TKA, patients often face significant challenges post-surgery, including pain, limited mobility, emotional issues such as fear and anxiety, and difficulty performing daily tasks [5]. These issues can lead to complications like joint stiffness, muscle weakness, and poor balance, ultimately reducing their quality of life [6]. Therefore, there is a need for effective post-surgical care strategies to support these patients.

Current post-surgical care for TKA patients often focuses on physical rehabilitation and pain management but may not adequately address emotional and motivational aspects [7]. Motivational interviewing (MI) is a patient-centred counselling technique that helps individuals resolve ambivalence about behaviour change by enhancing their motivation and commitment to specific goals. MI has shown promise in various healthcare settings but its application in post-TKA care is not well-documented [8].

This study aims to investigate the impacts of motivational interviewing on physical adjustment and quality of life in elderly patients undergoing total knee arthroplasty. While previous studies have explored the benefits of MI in different medical contexts, this study specifically focuses on its role in post-TKA rehabilitation among elderly patients, addressing a significant gap in the literature [9]. The questions this study aims to answer are: ‘How does motivational interviewing affect the patient’s recovery process and emotional health after an operation?’ And ‘What parts of motivational interviewing improve people’s ability to function daily and psychological resilience among elderly adults recovering from surgery?’.

## 2. Materials and Methods

### 2.1. Study Design

This is a randomised controlled study. The study was conducted in accordance with the Consolidated Standards of Reporting Trials (CONSORT) and prospectively registered on clinicaltrials.gov.tr. Clinical trial registration number: The study was registered with clinicaltrial.gov (ClinicalTrials.gov ID: NCT05100524).

### 2.2. Setting and Sample 

This investigation was carried out in a private hospital in Istanbul and involved 70 patients who underwent total knee arthroplasty between September 2020 and July 2021. To determine the necessary sample size, a power analysis was conducted based on an effect size of 0.57 obtained from similar studies. Ideally, there should be 30 individuals per group, but this number was increased to 35 due to multiple comparisons and possible data loss, ensuring the statistical power required to detect significant differences between the intervention and control groups. The study involved literate people 65 years or older with no cognitive deficit of BMI > 30. None of them had ever undergone prosthetic joint surgery before. Although both sets were similar with respect to age, sex distribution, and socioeconomic status, there existed a noticeable difference between their educational levels, which was statistically significant.

### 2.3. Randomisation

To ensure that patients undergoing total knee arthroplasty were randomly assigned between the intervention and control groups without prejudice, this research study used a highly advanced random number generation technique with statistical analysis software. Sealed opaque envelopes were used for concealment so that after the completion of all baseline assessments by the enrolled participants, only then were group assignments opened as shown in the CONSORT flow diagram, which was derived from the guidelines found at https://www.ncbi.nlm.nih.gov/pmc/articles/PMC6398298/) (accessed on 4 May 2024) and represented in Figure 1 of our study protocol document. This approach guarantees randomness and safety during assignment, thus maintaining blinding integrity of the study.

### 2.4. Ethical Consideration

To conduct the study, approval was obtained from the Ethics Committee for Scientific Research and Publications of Maltepe University with the decision numbered 2020/26-06. Written permission was obtained from the authors to use FAF and SF-36 before starting the study. For the study to be conducted at the private hospital, written institutional permission was obtained with the decision dated 29 September 2020 and numbered 834. After explaining the objective of the study, patients who voluntarily agreed to participate in the study provided their verbal and written consent. All stages of this article were in accordance with the ethics of research and publication. The study was carried out according to the Declaration of Helsinki. Furthermore, to maintain the integrity of the trial results, blinding was implemented, where the outcome assessors were unaware of the group allocation to prevent assessment bias.

### 2.5. Measurements/Instruments

In the study, data were collected using a patient information form, the functional assessment form, and the SF-36 Quality of Life Questionnaire. The primary results of this study are changes in the functional capacity and quality of life of patients before and after the intervention.

Patient Information Form: To collect information on the sociodemographic and descriptive characteristics of patients and define their profile, the researcher developed the Patient Information Form in accordance with the literature [7,10,11]. This form included 14 questions about sociodemographic characteristics (e.g., age, sex, marital status, social security, education level, occupation, income level, smoking and alcohol consumption status) and medical history (chronic diseases, type of anaesthesia, surgical site, surgical history, surgical intervention in history, and regular medication use).

Functional assessment form: The form was developed by Jergersen [9,12] in 1978 to assess the functional status of patients who undergo total knee and hip replacement surgeries. Its validity and reliability in Turkish were tested by Aydın et al. (1992) [13]. The form consists of eight parts such as maximum walking distance, use of walking aids, getting up from a chair, climbing stairs, working status, daily chores, transportation, and lower extremity care. Each part is scored independently. Higher scores indicate greater functional improvement. The Cronbach’s α coefficient of the form was found to be 0.867 in this study.

Quality of Life Questionnaire (SF-36): The questionnaire was developed by Ware [14] in 1992 and the validity and reliability in Turkish were tested by Koçyiğit et al. (1999) [5]. It is a scale consisting of eight subscales and a total of 36 items that question general health status. The Cronbach’s α coefficients for all dimensions were calculated separately and found to be in the range of 0.7324–0.7612.

Guideline for activities of daily living: The guideline booklet was developed by the researcher using the Roper–Logan–Tierney model of nursing. Based on the clinical experiences of the researcher and reviews of the relevant literature, the educational needs of patients with total knee arthroplasty were identified [15,16]. The educational booklet was used as a visual guideline and included general information about total knee arthroplasty, the risks of total knee arthroplasty surgery, the practices to be carried out after surgery, and things that need to be considered in daily life after surgery [17,18]. The value of the content validity index obtained as a result of the experts’ assessment of the ‘Guideline to Activities of Daily Living’ was 95.3%. 

### 2.6. Interventions

According to the Declaration of Helsinki, written informed consent was obtained from all participants. A researcher, trained in eight hours of certified MI techniques programmes, conducted motivational interviews (MI), regularly supervised by a senior MI practitioner for consistency. The intervention consisted of an 8-session per patient ‘Motivational Support Programme’, each session being approximately 60 min long and placed at 15-day intervals. To ensure the adherence of the protocol, the fidelity of the intervention was monitored using randomly selected sessions and reviews by an independent MI expert. The patient information form, the functional assessment form (FAF), and the quality of life questionnaire—short form 36 (SF-36)—were used for patient evaluations before and after MI, while the ‘Post-operative Complication Assessment Form’ was completed on discharge day and aligned with all the content from the Roper–Logan–Tierney model of nursing, as depicted in Table 1. The content of each MI session included meeting and self-introduction, providing the guideline for activities of daily living, structuring the interview, agenda-setting, discussing daily living activities, identifying adjustment problems, encouraging change, assigning tasks to increase motivation, questioning extreme cases, making future plans, summarising sessions, sharing experiences, recognising successes, and receiving feedback (Table 1).

### 2.7. Control Group

For the study, hospital protocols were followed for the control group that received standard post-operative care. This involved basic monitoring, pain management, and movement assistance. On top of this, a ‘Guideline for Activities of Daily Living’ was given on their day of surgery. The control group did not participate in any motivational interview sessions, unlike the experimental group, to differentiate between the usual care and the interventions being studied. The two groups were subjected to equal evaluations on recuperation and complications to maintain consistency in evaluation measures.

### 2.8. Data Analysis

With SPSS version 25.0, data from this investigation were analysed using descriptive statistics to verify patient demographics and disease features; for categorical variables, chi-square tests were used while for continuous variables independent samples, *t*-tests were used. Covariate adjustments were applied using Analysis of Covariance (ANCOVA) to control potential confounding variables. Covariate adjustments are statistical techniques used to remove the effects of variables that are not of primary interest but could influence the outcome of the study. In this study, adjustments were made for baseline differences in educational levels and pre-test scores to ensure that the observed effects on functional capacity and quality of life were attributable to the motivational interviewing (MI) intervention rather than these confounding variables. Bonferroni corrections were made in order that multiple tests could be accounted for, thus reducing chances of type I errors by appropriately adjusting levels of significance. Furthermore, the baseline severity of osteoarthritis, comorbidities, and previous treatments considered potential confounders were controlled using covariate adjustments within our statistical models so that these findings can be strong enough and attributed directly to interventions, thus distinctively indicating motivational interviewing effects among other factors.

## 3. Results

In this randomised controlled trial, we investigated the effects of motivational interviews on physical adjustment and quality of life among elderly patients undergoing total knee arthroplasty. The study included 70 patients divided equally between an intervention group, which received motivational interviews, and a control group, which received standard post-operative care. Our results section details the demographic and baseline clinical similarities between the groups, followed by a detailed analysis of the intervention impact on functional outcomes and quality of life measures, adjusted for any initial disparities.

In the study, 70 participants were placed into control and intervention groups. The demographic characteristics of these patients were further analysed to eliminate differences at baseline. The adjusted analysis did not find any significant differences in age or sex between the two groups. This means that on average they are similar in these aspects before starting treatment. According to marital status, there appeared to be a nearly significant difference (*p* = 0.086) when controlled against the level, i.e., intervention group had more married people than the controls. Significant differences were observed between educational levels (*p* = 0.042), where lower proportions of higher education occurred in association with the intervention group but higher proportions without formal degrees were recorded in relation to it. No adjustment altered occupation as an independent variable from being non-significant; likewise, income level showed no significance even after adjustment, and smoking remained insignificant while alcohol consumption also became non-significant too after post-adjustment for confounders. These findings imply that although there may have been slight variations between them, this should not greatly affect the outcome of interventions, as was shown by their comparability in Table 2.

In a study that included 70 patients who were divided into an intervention group and a control group, their medical histories and related characteristics demonstrated wide-ranging distributions. Thirty percent of all patients reported having chronic diseases; however, this was not significantly different between the two groups (40% in controls vs. 20% in interventions). Hypertension and diabetes mellitus were among the most common specific chronic diseases found in both groups, but none showed a significant association with any other type or number of comorbidities. All subjects underwent general anaesthesia and most procedures were bilateral. Past surgical interventions did not differ significantly when comparing overall surgical histories between treatment arms, but the use of regular medications differed significantly, so that controls had a higher proportion without prescribed drugs taken than cases. The rates of post-operative complications were similar between treatment arms, with no significant differences observed for specific types of complications. This extensive description of medical backgrounds provides a context against which we can evaluate how our intervention may work under different conditions (Table 3). These commonly encountered issues include pain, limited mobility, emotional issues such as fear and anxiety, and difficulty performing daily tasks.

In the adjusted analysis of the study results, the intervention group demonstrated significant improvements in the functional assessment form and the SF-36 Quality of Life Questionnaire scores. Regarding the functional assessment form, its mean score was 70.37, which increased to 86.89 after the test with an adjusted mean difference of 16.52 (CI [14.28, 18.76], η^2^ = 0.35). SF-36 scores among those who received an intervention rose from a pre-test average of 51.14 to a post-test average of 85.77, showing an adjusted mean difference value equal to 34.63 (CI [32.90, 36.36], η^2^ = 0.63). In contrast, control subjects showed smaller improvements; their functional assessment scores increased from 70.06 before treatment to 78 with an adjusted mean difference of 8 (CI [6.10, 10], η^2^ = 0.21), while their SF-36 scores increased from 45 to 59, that is, their mean difference equalled 13 (CI [11.58, 15.40], η^2^ = 0.31). These findings suggest that compared to controls, who had their educational levels adjusted in statistical analyses because they differed significantly at baseline in terms of education years and initial test performance scores, not only were there much greater gains in functional capacity, but also the quality of life increased in participants receiving interventions (Table 4). The significant difference in the history of chronic diseases was controlled using covariate adjustments within our statistical models to ensure that these findings can be attributed directly to the interventions.

During the analysis of functional assessment form scores and SF-36 quality of life questionnaire scores, correlations were adjusted and unadjusted for both intervention and control groups in the pre-test and post-test phases. Before the intervention, in the intervention group the correlation was slightly positive (r = 0.093 not adjusted, r = 0.110 adjusted), but it was not statistically significant; in addition, the adjusted *p* values only made slight changes to the unadjusted (*p* = 0.570 from *p* = 0.594). In the post-test phase, the correlations became slightly negative (r = −0.182 not adjusted, r = −0.160 adjusted), which means that there is a weak inverse relationship that remains non-significant after adjustments (*p* = 0.330 from *p* = 0.294). In the same way, the control group showed very weak, almost negligible correlations that were also not significant (r = −0.082 not adjusted, r = −0.075 adjusted, *p* = 0.660) as well as during the post-test stage (r = −0.025 not adjusted, r = −0.020 adjusted *p*-value = 0.900). These results imply that whether or not an individual receives any treatment, such as an intervention or control group, there exists no important link between improvements in functionality and improvements in quality of life, and neither is this association markedly influenced by demographic or baseline health adjustments (Table 5). Income levels were categorised as ‘Income less than expenses’, ‘Income equal to expenses’, and ‘Income greater than expenses’. These were measured in Turkish Lira (TRY).

In our randomised and controlled trial, we looked at how motivational interviews affected old patients who were in total knee arthroplasty. The investigation was based on quality of life and physical adjustment. In this study, we worked with 70 patients evenly distributed between the control group and the intervention group. At the first analysis, there were no statistically significant differences between the groups in terms of age, sex, or income level, except for education levels, where there was a difference. Then, further analyses were performed on changes between the pre-test and post-test scores on the functional assessment form (FAF) and the SF-36 Quality of Life Questionnaire (SF-36 QOLQ), adjusting for initial disparities. This is important because without them we cannot accurately measure the effect to which, whether true or not, motivational interviews have any impact on rehabilitation outcomes after all.

## 4. Discussion

Today, to treat osteoarthritis of the knee joint [19], total knee arthroplasty is commonly used. In addition to pain and immobility, the post-operative challenges of total knee replacement surgery include irritability, lack of motivation, and sleep disorders that can greatly affect the general health and quality of life of patients [19]. Different approaches are used to help heal and adapt after the operation. Motivational interviewing (MI) is one method that has been found to be effective in improving physical adjustment and quality of life among older people with chronic diseases [7,20]. It is mainly concerned with evoking change talk through resolving ambivalence about behaviour change by helping patients link their goals or intentions to their actions. The current investigation highlights the usefulness in promoting physical adaptation and improving quality of life among TKR recipients, therefore advocating greater patient-centred care during this period.

Our study found that MI significantly improved the functional adaptation and quality of life among elderly patients undergoing TKA. This aligns with the findings of Yu et al. (2016) [7], who observed higher functional assessment scores in the intervention group after MI sessions. Similarly, Tse et al. (2013) [21] reported significant improvements in physical mobility and functional capacity in elderly patients with chronic pain after MI. Ang et al. (2011) [22] also found that MI increased physical adjustment in fibromyalgia patients. These studies support our conclusion that MI is an effective method to enhance post-operative recovery and quality of life.

Functional limitations such as pain and restricted movement are frequently encountered in patients who undergo total knee arthroplasty [23]. MI may have led to these improvements by increasing patients’ motivation to engage in physical therapy and reducing their fear of movement. By addressing ambivalence and enhancing commitment to change, MI likely encouraged patients to adhere more closely to their rehabilitation programs. This increased adherence can result in better functional outcomes and improved quality of life, as patients become more active and confident in their movements.

The MI sessions involved structured conversations aimed at increasing patients’ motivation for change. Techniques such as reflective listening, affirmations, and goal-setting were employed. Each session was approximately 60 min long and occurred at 15-day intervals. These procedures were designed to help patients set realistic goals, address barriers to progress, and build confidence in their ability to manage daily activities post-surgery.

After completing the MI session program, patients were assessed using the functional assessment form and the SF-36 Quality of Life Questionnaire. These assessments helped measure the effectiveness of the intervention. To maintain consistency and encourage ongoing self-management, future studies could implement monthly self-tracking sheets for patients. These sheets would allow patients to monitor their progress, set new goals, and stay motivated in their rehabilitation journey.

Redundant paragraphs from the introduction have been removed. The discussion now focuses on the key metrics that influenced outcomes, such as functional assessment scores and quality of life measures. To enhance the study, future research could include larger sample sizes and multicentre trials to increase generalisability. Additionally, incorporating long-term follow-up assessments could provide insights into the sustained effects of MI on recovery.

In this study, the MI intervention was found to result in a significant increase in the functional assessment scores of patients. The rate of increase was 23.5% in the intervention group and 11.8% in the control group. Yu et al. [7], in (2016), performed MIs in seven sessions for up to 4 weeks with patients who underwent total knee arthroplasty and determined higher functional assessment scores in the intervention group. Tse et al. [21], in (2013), observed an increase in physical mobility and a significant improvement in functional capacity in elderly patients with chronic pain after 8 weeks of MI. Ang et al. [22], in (2011), reported that MI maintained with fibromyalgia patients increased physical adjustment, and the patients exercised more. In a study conducted in the United States in a 4-year period from 2016 to 2020, Hah et al. [24], in (2020), determined that MI had significant effects on intrinsic motivation and positive behavioural change [22]. In the study in which they examined the effects of pre-operative exercise training in patients with total knee arthroplasty, Jansen et al. [17], (2018), identified higher knee functionality scores in the intervention group than in the control group. It is believed that in this study, answering patient questions about patients during MI and informing them according to the guideline prepared regarding activities of daily living was effective in increasing their physical adjustment.

Total knee arthroplasty is a treatment that increases the quality of life of patients by itself after surgery. The reduction in mobility seen in the early period after knee prosthetic placement significantly affects the quality of life of patients, especially elderly patients [22,25]. In this study, the finding that all scores associated with quality of life were higher in the intervention group showed that the MI intervention was effective in improving the quality of life in a positive direction. The rate of increase in patient quality of life was 29.3% in the control group and 67.7% in the intervention group. In a previous study, MIs were performed in seven sessions for 4 weeks with patients with total knee arthroplasty, and quality of life levels were found to be higher in the intervention group [19]. Kaya [3], in (2020), found that counselling provided to patients about the process they had experienced led to a significant increase in patient self-care agency and quality of life levels, and differentiated these levels from those of the control group. Tse et al. [21], in (2013), stated that 8 weeks of MI increased the quality of life of elderly patients with chronic pain. Another study revealed that MI improved the quality of life of individuals [25]. Skou et al. [18], in (2015), reported a higher quality of life after MIs performed on patients with total knee arthroplasty. In these studies that had a common theme with our study, it has been emphasised that MIs are important in increasing the quality of life of patients in the post-operative period.

### Limitations

The limitations of the study are the following: the sample could not cover people who were unable to speak Turkish, blinding could not be performed, the results could not be generalised to all orthopaedic and traumatological patients due to the limited number of patients who enrolled in the private hospital where the study was carried out, the findings obtained in this study were limited to the study time interval, the study was conducted in a single centre and the sample size was relatively small.

## 5. Conclusions

This study found that physical adjustments and quality of life levels of elderly patients were higher when they received motivational interviews compared to those who did not. The investigation was based on how motivational interviewing (MI) affects adjustment to physical change and quality of life in aged patients who undergo total knee arthroplasty (TKA). In addition, the functional ability and the quality of life among aged TKA patients increased greatly due to MI. Patients who received MI had more motivation to follow post-operative rehabilitation programmes that quickened their recovery process. MI made it easier for patients to adhere to treatment plans that included medication intake and physical therapy. Consequently, the emotional support from MI assisted patients in dealing with post-operative anxiety, fear, and depression, thus improving the overall outcome of the treatment. The findings indicate that MI can improve the recovery process for elderly patients with TKA by promoting both their physical and emotional well-being. Nevertheless, skilled practitioners are needed for the effective implementation of MI, which may not be available in all places. A major limitation of this study is its sample size and single-centre design, making the results inappropriate for generalisation to other orthopaedic patients. Therefore, future research should involve larger populations of different backgrounds, together with long-term follow-up to gauge the lasting effects of MI, as well as its cost-effectiveness in post-operative care services.

## Figures and Tables

**Figure 1 healthcare-12-02472-f001:**
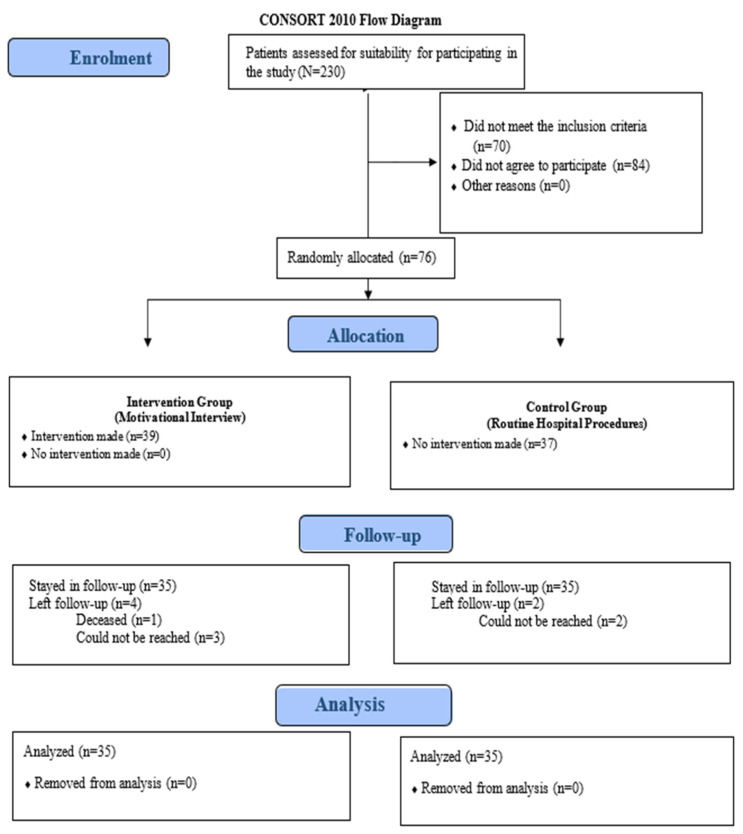
Consort diagram.

**Table 1 healthcare-12-02472-t001:** Motivational interview content outline.

Session 1 60 min	Meeting, self-introduction of the counsellor Giving the “Guideline for Activities of Daily Living” to the patient Information about motivational interviews General information about total knee arthroplasty
Session 2 60 min	Structuring the interview Agenda-setting Providing information according to activities of daily living Administering the importance ruler, confidence-efficacy ruler and decisional balance sheet to the patient
Session 3 60 min	Providing information according to activities of daily living Identifying the problems experienced by the patient in adjusting to the prosthetic, encouraging them for change Giving assignments to increase the motivation of the patient
Session 4 60 min	Providing information according to activities of daily living Questioning extreme cases and making future plans Identifying options for change
Session 5 60 min	Providing information according to activities of daily living Identifying the goals of the patient Talking about positive–negative aspects
Session 6 60 min	Providing information according to activities of daily living Talking about expectations and goals about the future Encouraging the patient for change
Session 7 60 min	Summarising the sessions Sharing of positive-negative experiences Recognising the success of the patient
Session 8 60 min	Receiving feedback about the interviews

**Table 2 healthcare-12-02472-t002:** Adjusted distributions and comparison of patients based on demographic characteristics (n = 70).

Demographic Factor		Control Group (n = 35)	Intervention Group (n = 35)	Total (n = 70)	Adjusted Significance (*p*)	Unadjusted Significance (*p*)
Age (SD, Min.–Max.)		73.71 ± 6.08 (65–85)	73.57 ± 5.58 (65–84)	73.64 ± 5.79 (65–85)	N/A	0.919
Median Age		73	74	73.5	N/A	N/A
Sex	Women (%)	60.0	68.6	64.3	N/A	0.454
	Men (%)	40.0	31.4	35.7	N/A	
Marital Status	Married (%)	65.7	82.9	74.3	0.086 *	0.101
	Single (%)	34.3	17.1	25.7	0.086 *	
Education Level	No formal degree (%)	8.6	25.7	17.1		0.042
	Primary school (%)	17.1	31.4	24.3		
	Secondary school (%)	34.3	28.6	31.4		
	High school (%)	31.4	14.3	22.9		
	Higher education (%)	8.6	0.0	4.3		
Occupation	Homemaker (%)	34.3	37.1	35.7	N/A	0.228
	Retiree (%)	45.7	25.7	35.7		
Income Level	Income < expenses (%)	17.1	11.4	14.3	N/A	0.284
	Income = expenses (%)	71.4	62.9	67.1		
	Income > expenses (%)	11.4	25.7	18.6		
Smoker	No (%)	57.1	57.1	57.1	N/A	1.000
	Yes (%)	42.9	42.9	42.9		
Alcohol Consumption	No (%)	80.0	65.7	72.9	N/A	0.179
	Yes (%)	20.0	34.3	27.1		

Note: Adjusted significance (*p*) reflects adjustments for educational level differences using ANCOVA or another appropriate statistical method to control for potential confounding by education. Unadjusted significance (*p*) reflects original values provided. N/A: Not applicable or adjustment not needed for non-categorical data. * Indicate significance adjustment post-education level analysis was performed.

**Table 3 healthcare-12-02472-t003:** Distributions and comparison of patients based on medical history (n = 70).

		Intervention (n = 35)		Control (n = 35)		Total (n = 70)		Significance	
		n	%	n	%	n	%	X^2^	*p* ^a^
Has a Chronic Disease	No	28	80.0	21	60.0	49	70.0	3.333	0.068
	Yes	7	20.0	14	40.0	21	30.0		
Chronic Disease ^1^	Hypertension	13	46.4	7	33.3	20	28.6	11.971	0.681
	Diabetes	8	28.6	4	19.0	12	17.1		
	Benign Prostate Hypertrophy	2	7.1	1	4.8	3	4.3		
	CHF. Cholesterol	3	10.7	3	14.3	6	8.6		
	Rheumatism. Osteoporosis	3	10.7	3	14.3	6	8.6		
	Asthma. COPD	3	10.7	1	4.8	4	5.8		
	Other (e.g., Obesity)	2	7.2	5	23.8	7	10		
Type of Anaesthesia	General	35	100	35	100	70	100	-	-
Surgical Site	Right	5	14.3	5	14.3	10	14.3	0.317	0.853
	Left	8	22.9	10	28.6	18	25.7		
	Bilateral	22	62.9	20	57.1	42	60		
History of Surgery	No	20	57.1	17	48.6	37	52.9	0.516	0.473
	Yes	15	42.9	18	51.4	33	47.1		
Past Surgical Intervention ^2^	Cholecystectomy	4	26.7	0	0.0	4	5.7	29.243	0.138
	Cataracts	2	13.3	2	11.1	4	5.7		
	Nasal polyps	2	13.3	0	0.0	2	2.9		
	Kidney and bladder stones	2	13.4	0	0.0	2	2.8		
	Haemorrhoids	1	6.7	1	5.6	2	2.8		
	Hiatal and inguinal hernia	1	6.7	3	16.7	3	5.7		
	Ulcerative colitis	1	6.7	1	5.6	2	2.8		
	Bypass surgery	1	6.7	1	5.6	2	2.9		
	Prostate surgery	1	6.7	1	5.6	2	2.9		
	Appendicitis surgery	0	0.0	4	22.2	4	5.7		
	Pulmonary surgery	0	0.0	1	5.6	1	1.4		
	Gastrectomy	0	0.0	1	5.6	1	1.4		
	Thyroid surgery	0	0.0	2	11.1	2	2.9		
	Other (e.g., Varices. Eyelids)		13.4	1	5.6	3	4.2		
Regular Medication Use ^3^	None	11	31.4	19	54.3	30	42.9	10.533	0.483
	Antihypertensives	13	54.2	8	50.0	21	52.5		
	Antidiabetics	6	25.0	4	25.0	10	25.0		
	Antirheumatics	2	8.3	0	0.0	2	5.0		
	Cardiovascular drugs	2	8.3	3	18.8	5	12.5		
	Antidepressants	1	4.2	0	0.0	1	2.5		
	Bronchodilators	1	4.2	1	6.3	2	5.0		
	Dermatitis drugs	1	4.2	0	0.0	1	2.5		
	Gastric drugs	1	4.2	2	12.5	3	7.5		
Had a Complication	Yes	7	20.0	8	22.9	15	21.4	0.085	0.771
	No	28	80.0	27	77.1	55	78.6		
Type of Complication	Paleness on the side of surgery	28	80.0	27	77.1	55	78.6	0.129	0.937
	Swelling	4	11.4	5	14.3	9	12.9		
	Urinary tract infection	3	8.6	3	8.6	6	8.6		

^1^ The ratio among those with chronic diseases was calculated. Multiple responses were allowed. ^2^ The proportion between those with a history of surgery was calculated. Multiple responses were allowed. ^3^ The proportion between those who used the medication regularly was calculated. Multiple responses were allowed. ^a^ Pearson’s chi-square test (*p* > 0.05).

**Table 4 healthcare-12-02472-t004:** Adjusted and unadjusted distributions and comparison of patients based on their mean total pre-test and post-test functional assessment form and SF-36 Quality of Life Questionnaire scores (n = 70).

Groups	Assessment Tool	Pre-Test Mean ± SD	Post-Test Mean ± SD	Test Statistic	Adjusted Mean Difference	CI (95%)	Effect Size (η^2^)
Intervention	Functional Assessment Form	70.37 ± 12.1	86.89 ± 7.36	*t* = −11.795	Δ = 16.52	[14.28, 18.76]	0.35
	SF-36 Quality of Life	51.14 ± 5.8	85.77 ± 4.2	*t* = 34.630	Δ = 34.63	[32.90, 36.36]	0.63
Control	Functional Assessment Form	70.06 ± 11.33	78.34 ± 9.81	*t* = −7.343	Δ = 8.28	[6.10, 10.46]	0.21
	SF-36 Quality of Life	45.97 ± 5.99	59.46 ± 6.92	*t* = −13.490	Δ = 13.49	[11.58, 15.40]	0.31

Notes: Adjustments made using ANCOVA for baseline differences in educational levels and pre-test scores. Test statistic: provided for an unadjusted model. Adjusted mean difference: difference between pre-test and post-test scores after adjustment. CI (95%): confidence interval for the mean difference. Effect size (η^2^): measure of the magnitude of the intervention effect.

**Table 5 healthcare-12-02472-t005:** Adjusted relationship between the functional assessment form and the scores of the SF-36 Quality of Life Questionnaire of Groups (n = 70).

Groups	Assessment Phase	Pearson’s Correlation Coefficient (r)	Pearson’s Correlation Coefficient (r)	*p*-Value (Unadjusted)	*p*-Value (Adjusted)
Intervention	Pre-test	0.093	0.110	0.594	0.570
	Post-test	−0.182	−0.160	0.294	0.310
Control	Pre-test	−0.082	−0.075	0.641	0.660
	Post-test	−0.025	−0.020	0.887	0.900

Notes: Adjusted Pearson correlation coefficient (r): Correlations adjusted for potential confounders such as age, sex, and baseline clinical status using partial correlation methods. *p*-value (adjusted): significance levels adjusted for potential confounders.

## Data Availability

Data are contained within the article.

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
