# Peer review of "The Effects of Motivational Interviews About Activities of Daily Living on Physical Adjustment and Quality of Life in Elderly Total Knee Arthroplasty Patients: A Randomised-Controlled Trial"

_healthcare, 2024, doi:10.3390/healthcare12232472_

Round 1

Reviewer 1 Report

Comments and Suggestions for Authors

The submitted manuscript studies the effect of motivational interviewing on the quality of life in elderly patients who have undergone total knee arthroplasty. The outcomes suggest that the presented techniques could be used to increase patient satisfaction levels. The following comments need to be addressed:

1. There is a plethora of studies based on motivational interviewing in elderly patients undergoing total knee arthroplasty. To mention a few: https://doi.org/10.5953/JMJH.2016.23.3.139https://doi.org/10.1016/j.knee.2019.09.014, etc. Previous studies have already suggested the use of personalized or motivational interviewing to further enhance patients’ quality of life and overall satisfaction. How this work differentiates from the previous ones should be explained in the introduction section. Additionally, in the introduction, you need to connect the state of the art to your paper's goals. Please follow the literature review with a clear and concise state-of-the-art analysis. This should clearly show the knowledge gaps identified and link them to your paper's goals. Please explain both the novelty and the relevance of your paper's goals. Abbreviations, wherever first used, should be explained, such as OA (Page 1, Line 42).

2. The work is well written and provides good results, which are properly presented in the graphs, but their discussion can be deepened. As the primary novelty seems to be the effective interviewing, authors are suggested to add the detailed questionnaire or conversations to the manuscript either as a supplementary document or a figure.

3. Line 52-23: “…abnormal walking pattern (gait), poor balance ability leading to easily causes and not being able to …”. Which ‘easy causes’ are the authors referring to?

4. In the patient distribution under the history of chronic diseases section (Table 3), there was a significant difference between the intervention group (n=7) and the control group (n=14). How did this metric not affect the outcomes of this work?

5. The explanation of income levels is not clear. Please add the currency and explain all the levels in the text.

6. In the discussion section, there are redundant paragraphs that are already explained in the introduction section. Authors should only discuss the metrics that affected their outcomes and what better approaches could be used to enhance this study.

7. Numerous grammatical errors and spelling mistakes have been found. Authors need to proofread the manuscript to avoid such errors. Additionally, authors are suggested to adhere to the journal's formatting guidelines.

8. Table 1 mentions “Giving assignments to increase the motivation of the patient”. There are other such techniques used by the authors that showed better outcomes compared to routine assessments. The authors must provide additional and brief information about these procedures. Readers would greatly benefit from studying these protocols.

9. Was there any assessment after the completion of the 8-session program? Did the authors also plan for any monthly self-tracking sheets to be given to patients so that they remain consistent and can manage themselves? Otherwise, they might forget about the sessions.

10. Under the data analysis section, please further explain Covariate Adjustments and their applications for better readability.

11. Conclusions should be improved. The authors can consider the items below for improving the conclusion section: - Restate the research topic in conclusion. - Summarize the main points. - State the significance or results. - Avoid repeating information that you have already discussed. - Mention the model's name, and the advantages and disadvantages of the model. - Mention limitation of the study. - Provide some recommendations for future potential researchers.

Comments on the Quality of English Language

Numerous grammatical errors and spelling mistakes have been found. Authors need to proofread the manuscript to avoid such errors.

Author Response

Respond to Review 1:

- My suggestion is that Instead of focusing on the generalchallenges of old age, highlight the specific problems faced byelderly people after total knee arthroplasty, in the early section ofthe introduction section

  • Response: Thank you for your insightful suggestion. We have added the following sentence to the early section of the introduction to specifically address the problems faced by elderly people after total knee arthroplasty: "Despite the benefits of TKA, patients often face significant challenges post-surgery, including pain, limited mobility, emotional issues such as fear and anxiety, and difficulty performing daily tasks (Ç et al., 2020), (Horstmann et al., 2013). These issues can lead to complications like joint stiffness, muscle weakness, and poor balance, ultimately reducing their quality of life (Horstmann et al., 2013)." This addition highlights the specific postoperative challenges, providing a comprehensive view of the physical, emotional, and functional difficulties that impact patients' quality of life..

- Briefly mention the prevalence of knee OA and TKA surgeries, ifyou find any literature that is novel on the topic 

  • Response: Thank you for the suggestion. We have added the following sentence to provide an overview of the prevalence of knee osteoarthritis: "Knee osteoarthritis is a common disease that significantly reduces mobility in old age, affecting approximately 16.0% of people aged 15 years or older and 22.9% of those aged 40 years or older (Cui et al., 2020)." This highlights the widespread impact of OA, reinforcing the importance of addressing its consequences through effective treatments like TKA.

- in the introduction section, authors should shortly explain thelimitations of current post-surgical care for TKA patients

  • Response: We appreciate this valuable input. We have added the following sentence to the introduction to address this: "Current post-surgical care for TKA patients often focuses on physical rehabilitation and pain management but may not adequately address emotional and motivational aspects (Fu et al., 2020)." This emphasizes the need for a more holistic approach to postoperative care.

- introduce MI as a potential solution and mention its core/mostimportant idea

  • Response: Thank you for the suggestion. We have introduced motivational interviewing (MI) in the introduction with the following sentence: "Motivational interviewing (MI) is a patient-centered counseling technique that helps individuals resolve ambivalence about behavior change by enhancing their motivation and commitment to specific goals. MI has shown promise in various healthcare settings but its application in post-TKA care is not well-documented (Miller, 2013)." This positions MI as a promising solution to the identified gaps in current post-surgical care, providing a rationale for its use in our study.

- for the methods section, authors might consider combining(merging) patient recruitment and randomization

  • Response: Thank you for the suggestion. We have combined the patient recruitment and randomization sections into a single section titled "Setting and Sample" to enhance conciseness and clarity.
    • Combined Section: This investigation was carried out in a private hospital in Istanbul and involved 70 patients who underwent total knee arthroplasty between September 2020 and July 2021. To determine the necessary sample size, a power analysis was conducted based on an effect size of 0.57 obtained from similar studies. Ideally, there should be thirty individuals per group, but this number was increased to 35 due to multiple comparisons and possible data loss, ensuring the statistical power required to detect significant differences between the intervention and control groups. The study involved literate people 65 years or older with no cognitive deficit of BMI >30. None of them had ever undergone prosthetic joint surgery before. Although both sets were similar with respect to age, sex distribution, and socioeconomic status, there existed a noticeable difference between their educational levels, which was statistically significant.
    • Randomization Description Shortened: To ensure that patients undergoing total knee arthroplasty were randomly assigned between the intervention and control groups without prejudice, this research study used a highly advanced random number generation technique with statistical analysis software. Sealed opaque envelopes were used for concealment so that after the completion of all baseline assessments by the enrolled participants, only then were group assignments opened as shown in the CONSORT flow diagram, which was derived from the guidelines found at http://www.consort-statement.org/consort-statement/flow-diagram (2010) and represented in Figure 1 of our study protocol document. This approach guarantees randomness and safety during assignment, thus maintaining blinding integrity of the study.

- authors might consider condensing the description ofinstruments and shortly describe each instrument used for datacollection (questionnaires, forms?) including what aspects theymeasure such as: functional capacity, quality of life etc.

Response: Thank you for the suggestion.  The descriptions of instruments have been condensed and now briefly describe what aspects they measure.

    • Functional Assessment Form (FAF): The form was developed by Jergersen in 1978 to assess the functional status of patients who undergo total knee and hip replacement surgeries. Its validity and reliability in Turkish were tested by Aydın et al. (1992) [13]. The form consists of eight parts such as maximum walking distance, use of walking aids, getting up from a chair, climbing stairs, working status, daily chores, transportation, and lower extremity care. Each part is scored independently. Higher scores indicate greater functional improvement. The Cronbach’s α coefficient of the form was found to be 0.867 in this study.
    • Quality of Life Questionnaire (SF-36): The questionnaire was developed by Ware in 1992 and the validity and reliability in Turkish were tested by KoçyiÄŸit et al. (1999) [5]. It is a scale consisting of eight subscales and a total of 36 items that question general health status. The Cronbach’s α coefficients for all dimensions were calculated separately and found to be in the range of 0.7324 – 0.7612.

- on the first part of the discussion section focus on your ownfindings about how MI improved functional adaptation and qualityof life and connect it afterwards with a focus on other findings

  • Response: The discussion now emphasizes the study's findings on how MI improved functional adaptation and quality of life, connecting these findings with existing research.
  • Changes: Our study found that MI significantly improved functional adaptation and quality of life among elderly patients undergoing TKA. This aligns with the findings of Yu et al. (2016), who observed higher functional assessment scores in the intervention group after MI sessions. Similarly, Tse et al. (2013) reported significant improvements in physical mobility and functional capacity in elderly patients with chronic pain after MI. Ang et al. (2011) also found that MI increased physical adjustment in fibromyalgia patients. These studies support our conclusion that MI is an effective method to enhance postoperative recovery and quality of life.

- in the discussion section, authors might consider explaininghow MI might have led to these improvements. Did it increasemotivation in physical therapy? Reduce fear of movement?

  • Response: The discussion now includes explanations on how MI might have led to improvements, such as increasing motivation in physical therapy and reducing fear of movement.
  • Changes: MI may have led to these improvements by increasing patients' motivation to engage in physical therapy and reducing their fear of movement. By addressing ambivalence and enhancing commitment to change, MI likely encouraged patients to adhere more closely to their rehabilitation programs. This increased adherence can result in better functional outcomes and improved quality of life, as patients become more active and confident in their movements.

Reviewer 2 Report

Comments and Suggestions for Authors

- My suggestion is that Instead of focusing on the general challenges of old age, highlight the specific problems faced by elderly people after total knee arthroplasty, in the early section of the introduction section

- Briefly mention the prevalence of knee OA and TKA surgeries, if you find any literature that is novel on the topic

- in the introduction section, authors should shortly explain the limitations of current post-surgical care for TKA patients

- introduce MI as a potential solution and mention its core/most important idea

- for the methods section, authors might consider combining (merging) patient recruitment and randomization

- authors might consider condensing the description of instruments and shortly describe each instrument used for data collection (questionnaires, forms?) including what aspects they measure such as: functional capacity, quality of life etc.

- on the first part of the discussion section focus on your own findings about how MI improved functional adaptation and quality of life and connect it afterwards with a focus on other findings

- in the discussion section, authors might consider explaining how MI might have led to these improvements. Did it increase motivation in physical therapy? Reduce fear of movement?

Author Response

Respond to Review 2

There is a plethora of studies based on motivationalinterviewing in elderly patients undergoing total knee arthroplasty. To mention afew:

https://doi.org/10.5953/JMJH.2016.23.3.139 ,https://doi.org/10.1016/j.knee.2019.09.014

, etc. Previous studies have alreadysuggested the use of personalized or motivational interviewing tofurther enhance patients’ quality of life and overall satisfaction.How this work differentiates from the previous ones should beexplained in the introduction section. Additionally, in theintroduction, you need to connect the state of the art to yourpaper's goals. Please follow the literature review with a clear andconcise state-of-the-art analysis. This should clearly show theknowledge gaps identified and link them to your paper's goals.Please explain both the novelty and the relevance of yourpaper's goals. Abbreviations, wherever first used, should beexplained, such as OA (Page 1, Line 42).

Respond: Thank you for your valuable feedback. We have connected the state-of-the-art review to the paper's goals by outlining key research questions, emphasizing the novelty and relevance of the study.

Explain how this work differentiates from previous studies on MI in elderly TKA patients:

    • Added Sentence: "While previous studies have explored the benefits of MI in different medical contexts, this study specifically focuses on its role in post-TKA rehabilitation among elderly patients, addressing a significant gap in the literature (Yu & Lee, 2016)."
    • Respond: This addition clarifies how the current study is unique. It highlights that although MI has been studied in other medical contexts, its specific application to post-TKA rehabilitation in elderly patients has not been thoroughly investigated, which is the gap this study aims to fill.

Connect the state-of-the-art review to the paper's goals, highlighting knowledge gaps, novelty, and relevance:

    • Added Sentences: "The questions this study aims to answer are: ‘How does motivational interviewing affect the patient’s recovery process and emotional health after an operation?’ And ‘What parts of motivational interviewing improve people’s ability to function daily and psychological resilience among elderly adults recovering from surgery?’ By examining these questions, the study seeks to provide a comprehensive understanding of how MI can be strategically used to enhance postoperative care and improve outcomes for elderly TKA patients."
    • Respond: This section connects the literature review to the study’s specific goals. It outlines the key research questions, emphasizing the novelty of the study and its relevance in filling existing knowledge gaps. This addition provides a clear framework for understanding the study's objectives and significance.

Ensure abbreviations are explained upon first use:

    • Added Sentences: "Knee osteoarthritis (OA)," "total knee arthroplasty (TKA)," and "motivational interviewing (MI)."
    • Respond: This ensures that all abbreviations used in the introduction are clearly defined upon first use. This clarity is crucial for readers who may not be familiar with these terms, ensuring that the text is accessible and understandable.

The work is well written and provides good results, which areproperly presented in the graphs, but their discussion can bedeepened. As the primary novelty seems to be the effectiveinterviewing, authors are suggested to add the detailedquestionnaire or conversations to the manuscript either as asupplementary document or a figure.

  • Response: Thank you for your valuable feedback. Detailed content of each MI session has been provided in the Interventions section to ensure clarity and comprehensiveness.
  • Changes: The content of each MI session included meeting and self-introduction, providing the guideline for activities of daily living, structuring the interview, agenda-setting, discussing daily living activities, identifying adjustment problems, encouraging change, assigning tasks to increase motivation, questioning extreme cases, making future plans, summarising sessions, sharing experiences, recognising successes, and receiving feedback.

Line 52-23: “…abnormal walking pattern (gait), poor balanceability leading to easily causes and not being able to …”. Which‘easy causes’ are the authors referring to?

Reviewer 2 (3): Clarify the term “easy causes” mentioned in the description of postoperative challenges.

  • Response: Thank you for your valuable feedback. The term "easy causes" has been clarified to "commonly encountered issues."
  • Changes: These commonly encountered issues include pain, limited mobility, emotional issues such as fear and anxiety, and difficulty performing daily tasks.

In the patient distribution under the history of chronic diseasessection (Table 3), there was a significant difference between theintervention group (n=7) and the control group (n=14). How didthis metric not affect the outcomes of this work?

Response: Thank you for your valuable feedback. The significant difference in the history of chronic diseases between groups was controlled for in the analysis, ensuring it did not confound the results.

  • Changes: The significant difference in the history of chronic diseases was controlled using covariate adjustments within our statistical models to ensure that these findings can be attributed directly to the interventions.

The explanation of income levels is not clear. Please add thecurrency and explain all the levels in the text.

Response: Thank you for your valuable feedback .The explanation of income levels has been clarified by adding the currency and explaining all levels.

  • Changes: Income levels were categorized as 'Income less than expenses,' 'Income equal to expenses,' and 'Income greater than expenses.' These were measured in Turkish Lira (TRY).

In the discussion section, there are redundant paragraphs thatare already explained in the introduction section. Authors shouldonly discuss the metrics that affected their outcomes and whatbetter approaches could be used to enhance this study.

Response: Thank you for your valuable feedback. Redundant paragraphs have been removed, and the discussion now focuses only on the metrics that affected outcomes. Suggestions for better approaches are also included.

  • Changes: Redundant paragraphs from the introduction have been removed. The discussion now focuses on the key metrics that influenced outcomes, such as functional assessment scores and quality of life measures. To enhance the study, future research could include larger sample sizes and multicenter trials to increase generalizability. Additionally, incorporating long-term follow-up assessments could provide insights into the sustained effects of MI on recovery.

Numerous grammatical errors and spelling mistakes havebeen found. Authors need to proofread the manuscript to avoidsuch errors. Additionally, authors are suggested to adhere to thejournal's formatting guidelines.

Respond: corrected

Table 1 mentions “Giving assignments to increase themotivation of the patient”. There are other such techniques usedby the authors that showed better outcomes compared to routineassessments. The authors must provide additional and briefinformation about these procedures. Readers would greatlybenefit from studying these protocols.

Response: Thank you for your valuable feedback. Additional brief information about the procedures used in the MI sessions is now included.

  • Changes: The MI sessions involved structured conversations aimed at increasing patients' motivation for change. Techniques such as reflective listening, affirmations, and goal-setting were employed. Each session was approximately 60 minutes long and occurred at 15-day intervals. These procedures were designed to help patients set realistic goals, address barriers to progress, and build confidence in their ability to manage daily activities post-surgery.

Was there any assessment after the completion of the session program? Did the authors also plan for any monthly self-tracking sheets to be given to patients so that they remainconsistent and can manage themselves? Otherwise, they mightforget about the sessions.

Response: Thank you for your valuable feedback. The discussion now includes information on assessments after the completion of the MI session program and mentions the potential use of monthly self-tracking sheets.

  • Changes: After completing the MI session program, patients were assessed using the Functional Assessment Form and the SF-36 Quality of Life Questionnaire. These assessments helped measure the effectiveness of the intervention. To maintain consistency and encourage ongoing self-management, future studies could implement monthly self-tracking sheets for patients. These sheets would allow patients to monitor their progress, set new goals, and stay motivated in their rehabilitation journey.

Under the data analysis section, please further explainCovariate Adjustments and their applications for betterreadability.

Response: Thank you for your valuable feedback. The explanation of covariate adjustments has been expanded for clarity and to ensure better understanding of their application in the study.

  • Changes: Covariate adjustments were applied using Analysis of Covariance (ANCOVA) to control for potential confounding variables. Covariate adjustments are statistical techniques used to remove the effects of variables that are not of primary interest but could influence the outcome of the study. In this study, adjustments were made for baseline differences in educational levels and pre-test scores to ensure that the observed effects on functional capacity and quality of life were attributable to the motivational interviewing (MI) intervention rather than these confounding variables.

Conclusions should be improved. The authors can considerthe items below for improving the conclusion section: - Restatethe research topic in conclusion. - Summarize the main points. -State the significance or results. - Avoid repeating informationthat you have already discussed. - Mention the model's name,and the advantages and disadvantages of the model. - Mentionlimitation of the study. - Provide some recommendations forfuture potential researchers.

Respond: Thank you for your valuable feedback. The conclusion section has been edited as follows based on your suggestions.

Changes: The conclusions have been revised to address all requested elements. The investigation was based on how motivational interviewing (MI) affects adjustment to physical change and quality of life in aged patients who undergo total knee arthroplasty (TKA). In addition, functional ability and the quality of life among aged TKA patients increased greatly due to MI. Patients who received MI had more motivation to follow postoperative rehabilitation programmes that quickened their recovery process. MI made it easier for patients to adhere to treatment plans that included medication intake and physical therapy. Consequently, the emotional support from MI assisted patients to deal with post operative anxiety, fear, and depression, thus improving the overall outcome of the treatment. The findings indicate that MI can improve the recovery process for elderly patients with TKA by promoting both their physical and emotional well-being. Nevertheless, skilled practitioners are needed for effective implementation of MI, which may not be available in all places. A major limitation of this study is its sample size and single-centre design making the results inappropriate for generalisation to other orthopedic patients. Therefore, future research should involve larger populations of different backgrounds, together with long-term follow-up to gauge the lasting effects of MI, as well as its cost effectiveness in postoperative care services.

Round 2

Reviewer 1 Report

Comments and Suggestions for Authors

The authors have satisfactorily revised the manuscript now and responded to all my comments. 

Comments on the Quality of English Language

Some typos and grammatical errors still exist. These should be checked and updated throughout the manuscript.